# Co-Doped CeO_2_/Activated C Nanocomposite Functionalized with Ionic Liquid for Colorimetric Biosensing of H_2_O_2_ via Peroxidase Mimicking

**DOI:** 10.3390/molecules28083325

**Published:** 2023-04-09

**Authors:** Abdul Khaliq, Ruqia Nazir, Muslim Khan, Abdur Rahim, Muhammad Asad, Mohibullah Shah, Mansoor Khan, Riaz Ullah, Essam A. Ali, Ajmir Khan, Umar Nishan

**Affiliations:** 1Department of Chemistry, Kohat University of Science and Technology, Kohat 26000, Pakistan; 2Department of Chemistry, COMSATS University Islamabad, Park Road, Islamabad 45550, Pakistan; 3Department of Biochemistry, Bahauddin Zakariya University, Multan 66000, Pakistan; 4Department of Pharmacognosy, College of Pharmacy, King Saud University, Riyadh 11451, Saudi Arabia; 5Department of Pharmaceutical Chemistry, College of Pharmacy, King Saud University, Riyadh 11451, Saudi Arabia; 6School of Packaging, 448 Wilson Road, Michigan State University, East Lansing, MI 48824, USA

**Keywords:** H_2_O_2_, Co-doped CeO_2_/activated C nanocomposite, ionic liquid, peroxidase mimic, colorimetric sensor

## Abstract

Hydrogen peroxide acts as a byproduct of oxidative metabolism, and oxidative stress caused by its excess amount, causes different types of cancer. Thus, fast and cost-friendly analytical methods need to be developed for H_2_O_2_. Ionic liquid (IL)-coated cobalt (Co)-doped cerium oxide (CeO_2_)/activated carbon (C) nanocomposite has been used to assess the peroxidase-like activity for the colorimetric detection of H_2_O_2_. Both activated C and IL have a synergistic effect on the electrical conductivity of the nanocomposites to catalyze the oxidation of 3,3′,5,5′-tetramethylbenzidine (TMB). The Co-doped CeO_2_/activated C nanocomposite has been synthesized by the co-precipitation method and characterized by UV–Vis spectrophotometry, FTIR, SEM, EDX, Raman spectroscopy, and XRD. The prepared nanocomposite was functionalized with IL to avoid agglomeration. H_2_O_2_ concentration, incubation time, pH, TMB concentration, and quantity of the capped nanocomposite were tuned. The proposed sensing probe gave a limit of detection of 1.3 × 10^−8^ M, a limit of quantification of 1.4 × 10^−8^ M, and an R^2^ of 0.999. The sensor gave a colorimetric response within 2 min at pH 6 at room temperature. The co-existing species did not show any interference during the sensing probe. The proposed sensor showed high sensitivity and selectivity and was used to detect H_2_O_2_ in cancer patients’ urine samples.

## 1. Introduction

Hydrogen peroxide is one of the most important analytes produced in many oxidative metabolic reactions as a byproduct and is catalytically degraded by natural oxidases in living organisms under ambient conditions of temperature, pH, etc. [1]. It is also produced in several fatal diseases such as hepatitis, atherosclerosis, and severe infections [2]. Hydrogen peroxide has a key role in organic synthesis, food security, bioanalysis, and environmental protection. The concentration of H_2_O_2_ in biological systems is regulated by the enzyme peroxidases, which have been released by human kidneys and red blood cells. The abnormal production of H_2_O_2_ in an individual body is indicated by respiratory disorders, inflammation, diabetes, aging, and oxidative stress, which induce the denaturation of proteins and damage DNA. Ultimately, this DNA damage leads to the occurrence of different types of cancer [3]. The evaluation, as well as the monitoring of H_2_O_2_, are also essential in food production, chemical, biological, clinical, pulp, paper bleaching, and sterilization processes [4]. Thus, it is essential to check the concentration of hydrogen peroxide because it is one of the most important analytes attracting the attention of researchers [5].

In recent years, many approaches, such as the chromatographic [6], spectrophotometry [7], chemiluminescence [8], and electrochemical methods [9], have been reported for the determination of hydrogen peroxide. Requirements for expensive instrumentation, skilled operators, and high maintenance costs, make it tedious and reduce its feasibility in resource-limited laboratories [10]. Enzymatic sensing methods are also used, but their high cost, low stability, difficulty in transportation, delicate handling, and low shelf life make them undesirable [11]. Compared to the other experimental techniques, colorimetric biosensors based on enzyme mimicry are simple, cheap, robust, highly sensitive, and selective for onsite hydrogen peroxide detection, and the color of the reaction can be easily observed with the naked eye [12,13].

Among the metal oxides, cerium oxide is one of the reactive rare earth metal oxides that has gained a lot of interest. The use of CeO_2_ as a colorimetric sensing platform for H_2_O_2_ has been reported by Cheng et al. [14]. However, the reported work limit of detection is higher than the desired value for H_2_O_2_ sensing. Remani and Binitha [15] used cobalt doping on nanoceria to improve its oxidizing nature. Furthermore, Rauf et al. [16] reported on the use of activated carbon as a cost-effective and recyclable peroxidase mimics in the construction of colorimetric sensors for biomedical applications. The decoration of doped metal oxides with activated carbon also increases the surface area and the electrical conductivity of nanocomposites, which are the basic requirements for nanocomposites to be used as sensors and biosensors [17]. Ionic liquid (IL) is an excellent medium for the applications of nanocomposites due to its good physiochemical properties, great biocompatibility, low vapor pressure, and non-volatility [18]. It can be used as a capping agent to increase the stability and electrical conductivity of the nanomaterials and prevent their aggregation [19]. After a thorough review of the literature, we strategized to explore the synergistic effect of Co, CeO_2,_ activated carbon, and ionic liquid for the sensing of H_2_O_2_.

In the present research work, we reported a nanocomposite in which cerium oxide-doped, cobalt decorated with activated carbon (Co-CeO_2_/C) was synthesized. Furthermore, Co-CeO_2_/C were functionalized with an ionic liquid-like 1-Butyl-3-methylimidazolium tetrafluoroborate by coating their surfaces physically in order to increase their deagglomeration, electrical conductivity, stability, and catalytic activity [20]. Then the nanocomposite was analyzed using a number of methods, including UV–Visible spectroscopy, EDX, Raman spectroscopy, SEM, X-ray diffraction, and FTIR. To attain the best effectiveness of the IL-capped Co/CeO_2_/C nanocomposite, various parameters, such as (a) amount of nanocomposite (b) TMB concentration, (c) pH (d) H_2_O_2_ concentration and (e) incubation period, have been optimized. The fabricated material was then successfully applied as a biosensing probe for the colorimetric analysis of hydrogen peroxide in the urine samples of cancer patients.

## 2. Results and Discussion

### 2.1. Characterization of Cobalt-Doped Cerium Oxide/Activated Carbon Nanocomposite

#### 2.1.1. FTIR Spectra of the Nanocomposite

Figure 1A,B show the Fourier transform infrared spectra of the synthesized nanocomposite. The peaks below the range of 700 cm^−1^, including 617, 595, 586, 564, 555, and 549 cm^−1^, indicate the CeO stretching mode and it confirms the structure of CeO_2_ [21,22]. The peak at 846 cm^−1^ is related to O-C-O stretching vibration. While the peaks obtained at 961 cm^−1^, and 1090 cm^−1^ belong to C-O stretching vibration, CO_2_ asymmetric stretching vibration, and CO_3_^−2^ bending vibration. Furthermore, the appearance of a peak at 1363 cm^−1^ indicates an N-O stretching band of oxygen present in cerium with nitrogen. In addition, the absorption peak at 1636 cm^−1^ corresponds to the bending vibration of H-O-H [22]. The band at 2438 cm^−1^ shows the presence of dissolved CO_2_ in the sample, and the absorption peaks at 3227 cm^−1^ and 3698 cm^−1^ were due to the stretching vibration of OH or due to the physical absorption of H_2_O and the surface OH group [23]. The weak band at 2037 cm^−1^ shows the presence of cobalt (Co) bonded to a metallic fraction of cobalt [24]. The peak at 1495 cm^−1^ correlated to C-H symmetric and asymmetric bending vibration [25]. From this spectral evidence, it was confirmed that the synthesized nanomaterials were activated carbon-decorated Co-doped CeO_2_ nanocomposites.

#### 2.1.2. XRD Pattern of Synthesized Nanostructures

The XRD pattern of cobalt-doped cerium oxide/activated carbon is shown in Figure 2. The XRD analysis between 20–80° confirms the crystalline structure of the prepared nanocomposite. The different diffraction peaks located at 28°, 33°, 47°, 56°, 69°, and 76.76° can be indexed into 111, 200, 220, 311, 400, and 331, respectively, showing that the prepared nanocomposite has a face-centered cubic structure. There is strong agreement between all of the peaks and the Joint Committee on Powder Diffraction Standards (JCPDS no: 34–0394). When we compared the obtained spectrum with the literature, we observed the broadening of the peaks that can be attributed to the presence of doping materials. Similar peak broadening has also been reported in the literature [26]. The prominent peak corresponds to 110 millers indexed implying that major orientation occurs at 110 indices.

The Scherrer equation was used to compute the average particle size of cobalt-doped cerium oxide/C for the peak with the highest intensity on the face-centered cube.

D = Kλ/βcosθ

Where D = the average crystal size

K = Sherrer constant (K = 0.94)

Λ = wavelength of X-ray source (CuKα = 1.54184 Å)

Β = represents full width at half maximum (FWHM) of the diffraction peak expressed in radian

θ = The diffraction angle at a peak maximum

The average crystal size for the synthesized nanocomposite was determined to be 3.81 nm.

#### 2.1.3. SEM Study of the Synthesized Nanostructures

The surface structure of activated carbon-decorated cobalt-doped cerium oxide was studied by using an electron microscope as shown in Figure 3. SEM is applied at an accelerated voltage of 20 kV to characterize the particle size, shape, and surface morphology [27]. The SEM images show that the carbon-decorated cobalt-doped cerium oxide nanocomposite has an agglomerated cluster structure of granular shape. The surface of Co-CeO_2_/C has been found to possess voids of different sizes, possibly due to the maximum adsorption of TMB over the large surface of the synthesized nanomaterials [28].

#### 2.1.4. EDX Analysis of the Synthesized Co/CeO_2_/C Nanocomposite

The EDX analysis of carbon-decorated cobalt-doped cerium oxide is shown in Figure 4. This analysis confirms the presence of carbon, oxygen, cobalt, and cerium in the sample of the prepared nanocomposite [29]. The carbon, oxygen, cobalt, and cerium contents by weight % were found to be 13.90, 23.95, 0.20, and 61.94, respectively, as shown in Table 1.

#### 2.1.5. Raman Spectrum of the Synthesized Co/CeO_2_/C Nanocomposite

Raman spectrometry is an important instrumental technique used for analyzing structural defects, diverse phase purity, and some disorders such as a deficiency of oxygen in the synthesized nanocomposites. The distinctive peak at 450 cm^−1^, which relates to the F_2_g symmetric stretching vibration state in Ce-O_8_ of CeO_2_, indicates the cubic fluorite structure of CeO_2_, which is produced by the doping of cobalt in CeO_2_/C and causes oxygen defects in cerium oxide [30]. The result shown in Figure 5 is that the doping of cobalt in cerium oxide can transfer the band to a lower region of 450 cm^−1^ from 465 cm^−1^ because the oxygen vacancies are being created in cerium oxide [31]. This further enhances the entry of the cobalt cation into the crystal structure of CeO_2_. The particle size also affects the position of the band and it has also been observed that the shifting of the band to the lower regions is due to the reduction in particle size [32]. The band at 846 cm^−1^ corresponds to the V(O–O) stretching vibration of peroxo species formed on CeO_2_, while the bands at 1160 cm^–1^ were attributed to second-order longitudinal optical (2LO) mode [33]. The peaks at 110, 118, 237, and 601 cm^−1^ indicate the spinal structure of Co_3_O_4._ These peaks do not appear clearly in Figure 5. Therefore, Appendix A (Appendix A) has been provided, which clearly shows the peaks at 110, 118, 237, and 601 cm^−1^. These Raman modes are most likely associated with the tetrahedral site and the octahedral oxygen moment [34].

### 2.2. Colorimetric Sensing of H_2_O_2_

Hydrogen peroxide was determined through a colorimetric approach. Cobalt-doped cerium oxide/activated carbon nanocomposite was used as a sensing platform. Figure 6 shows the UV–Vis spectra and optical change in the proposed sensor. By adding hydrogen peroxide, the sensor can efficiently detect it by changing the colorless TMB solution into a blue–green product. Additionally, the TMB has been subsequently oxidized by the OH radicals produced by H_2_O_2_ adsorbed on the surface of the capped Co/CeO_2_/C nanocomposite. Figure 6, curve A, shows (A) the absence of H_2_O_2_ and (B) the presence of H_2_O_2_. The UV−Vis spectra peak at 652 nm also confirms this change.

### 2.3. Mechanism for Sensing of H_2_O_2_

Experimentally, the catalytic activity of IL/cobalt-doped cerium oxide/activated carbon nanocomposite as a peroxidase was evaluated from the oxidation of TMB by H_2_O_2_. The absorbance peak of the resultant bluish-green color product (oxidized TMB) was recorded at 653 nm by using a UV–Vis spectrophotometer. The reaction mechanism is given as: IL/capped cobalt-doped cerium oxide/activated carbon nanocomposite absorb light energy (photons). The band gap energy of the nanocomposite is equal to the energy of the absorbed photon. The electron is excited from the valence band to the conduction band because electrons and H^+^ are produced due to the absorption of photons. Actually, the initiation of the reaction is performed by the dissociation of H_2_O_2_, not TMB, assisted by the Co-doped CeO_2_/AC nanocomposite by transferring electrons. CeO_2_ has a wide band gap of about 3.0 eV, as reported by Sharma et al. [35]. Upon doping with Co and activated carbon the band gap is reduced. Habib et al. [36] reported the reduction of the band gap energy of CeO_2_ with chromium (Cr). These excited electrons can be scavenged by H_2_O_2_ to create OH^•^ radicals and OH^−^. H_2_O_2_ will also prevent H^+^ and electron recombination. The peroxo complex is formed between cerium oxide and H_2_O_2_ when the hydrogen peroxide adsorbed on the surface of cobalt-doped cerium oxide/activated carbon nanocomposite is functionalized with ionic liquid, which further increases the dissociation of H_2_O_2_ by the absorption of electrons. It enhances the oxidation of TMB, leading to a bluish-green product as shown in Figure 1 [37,38]. Here, the electrical conductivity refers to the addition of an electron for the dissociation of H_2_O_2_ and oxidation of TMB assisted by the Co-doped CeO_2_/AC nanocomposite by transferring electrons.

### 2.4. Optimization of Experimental Parameters

#### 2.4.1. Effect of Capped Nanocomposite

Ionic liquids play a significant role in the stabilization of metal nanocomposites because they prevent undesired agglomeration of the nanocomposites by exerting electrostatic and stearic stabilization and so improving the catalytic activities. Previously, for the detection of hydrogen peroxide, different nanomaterials were employed as a colorimetric sensing platform but they gave the result with a time delay, were not comprehensive, and lacked the necessary details [39,40]. Herein the current work, we have successfully synthesized the cobalt-doped cerium oxide/C nanocomposite capped with an ionic liquid to avoid agglomeration and it was also employed in real samples in order to fill the gap. Because of the good dispersion power, strong π−π stretching, and weak interactions with the substrate, ionic liquids play a key role in increasing catalytic power. Further, the acidic proton of the cationic part of the ionic liquid is involved in the decomposition of H_2_O_2_ by providing an OH radical, which can easily oxidize TMB. In our research work, Figure 7 shows the IL/coated nanocomposite optimization. The amount of ionic liquid capped cobalt-doped cerium oxide/activated carbon was changed from 20–80 μL. By using 50 μL IL/capped nanocomposite, the visible color change and the prominent UV peaks appeared. Initially, by using a lesser amount of capped nanocomposite, there was insufficient OH radical to completely oxidize the entire TMB. It was observed that by increasing the amount of capped nanocomposite the color of the reacting substances gradually changed due to the oxidation of TMB. However, beyond the optimum concentration of the nanocomposite, the color started to get faint, as can be seen in Figure 7 tubes E-G. This indicates that when the amount of capped nanocomposite exceeds the threshold the excess will be dispersed and no agglomeration will occur in the reaction medium, which is reliable and according to the reported literature [28].

#### 2.4.2. Time Optimization

Figure 8 shows the effect of time on sensor response from 30–210 s. The suggested sensor performs best in 120 s because the color totally changes from colorless to blue–green within 120 s and produces the highest peak levels. After 120 s, there was no color change, therefore, for the proposed sensor, 120 s was considered the optimum time. All other experimental tests were performed at this time. The already reported sensor shows the best response at 25–30 min [40,41] while our proposed sensor gives a rapid response compared to the already reported sensor.

#### 2.4.3. pH Optimization

While assessing the effectiveness of biosensors, pH measurement is indispensable. The pH analysis was carried out in the range of 3–11 by utilizing HCl as well as NaOH solutions to optimize the pH for the suggested sensor. The best response was observed at pH 6.0 within 2 min with a bluish-green color as shown in Figure 9. The literature review also indicates that 7.4 is the ideal pH for the proposed sensor [42].

#### 2.4.4. Effect of TMB Concentration

The effectiveness of TMB concentration on a biosensor is indicated in Figure 10. The absorbance increased very rapidly till point D as the TMB solution concentration increased. The highest colorimetric color change occurs at an 8 mM TMB concentration. The color becomes faint by increasing the concentration of TMB further because all the other available components are consumed. A further increase in TMB concentration results in a mixture of oxidized and reduced TMB. Since oxidized TMB is blue–green in color and unreacted TMB is colorless it results in reducing the color absorbance. Therefore, an 8 mM concentrated solution of TMB was selected as the optimum range for visible colorimetric color change. While in the literature, 12 mM [42], and 18 mM [43] concentrations of TMB were also reported. There is no response between Co-CeO_2_/C/IL and H_2_O_2_ in the absence of TMB. This demonstrates that Co-CeO_2_/C/IL has peroxidase-like activity comparable to that of other metal-based peroxidase mimics.

#### 2.4.5. Colorimetric Determination of H_2_O_2_

By studying the research to identify the best experimental conditions based on the peroxidase-like activity of Co/CeO_2_/C/IL coated with ionic liquid, a relatively easy UV colorimetric approach was used to measure H_2_O_2_. A very helpful and selective colorimetric approach was used, keeping in mind the color change in relation to the quantitative measurement of H_2_O_2_ indicating the relationship between the absorption intensity at 652 nm and H_2_O_2_ concentration. The sensitive application of the proposed sensor was investigated by using H_2_O_2_ solutions of different concentrations. Figure 11A indicates that the response and peak intensity of the proposed sensor are low at lower concentrations of H_2_O_2_ and linearly increase with increasing concentrations of H_2_O_2_ solution. This technique of biosensor can determine H_2_O_2_ concentration with a linear range of 1 × 10^−9^ M to 3.6 × 10^−7^ M with an R^2^ value of 0.999, an intercept 0.13699 absorbance (a.u.), a slope of 1,139,000 absorbance/M and a standard error of 0.00161 in the inset Figure 11B. The limit of detection (LOD) and limit of quantification (LOQ) of the fabricated platform are 1.3 × 10^−8^ M and 1.4 × 10^−8^ M, respectively. The minor changes in analyte concentration can show a linear response with respect to the nanocomposite sensor. The suggested sensor is compared to the already published literature [44], indicating that our designed sensor is very sensitive and shows a very good response to very low concentrations of H_2_O_2_. It is clear from this research that this technique provides a very helpful way to measure the concentration of H_2_O_2_ by taking UV spectra at 652 nm with a blue–green color that is absolutely distinguishable by the naked eye. In addition, the absorption of H_2_O_2_ on the surface of the nanocomposite produces OH radicals, which then take part in the oxidation of TMB to produce a bluish-green substance [45].

### 2.5. Interference Studies

In the presence of potentially interfering substances such as methanol, ethanol, Ca^+2^, urea, and uric acid, the suggested sensor consisting of Co/CeO_2_/C coated with ionic liquid as a peroxidase mimic for the detection of H_2_O_2_ was examined for selectivity. The productivity of the developed sensor greatly depends on the interference study in the biomedical field toward clinical diagnosis. In the presence of interference chemicals, the biosensing efficiency of the proposed sensor was investigated in terms of absorbing responses as shown in Figure 12. It is obvious from the figure that by increasing the concentration of H_2_O_2_ the spectra of the bluish-green color product at 652 mm increase even in the presence of interference chemicals of the same concentration [40]. The experiment was repeated under the same conditions with the interfering chemicals’ concentrations raised up to 4 folds (1.0 × 10^−6^ M) compared to hydrogen peroxide (3.6 × 10^−7^ M), but no visible change in color or absorbance was seen. Hence, this assay further supported the conclusions that the suggested sensor is extremely sensitive and selective for the detection of hydrogen peroxide in the presence of the interfering material. These findings are in accordance with the published research [42].

### 2.6. Analysis of Real Samples

To test the suggested biosensor’s practical applicability, cancer patients’ urine samples were used The literature survey reveals that very few recently applied the proposed sensor to real samples [20]. As indicated in Table 2, various H_2_O_2_ concentrations, including 17, 82, and 244 nM, were spiked into urine samples taken from cancer patients at the IRNUM hospital in Peshawar, Khyber Pakhtunkhwa, Pakistan. By using H_2_O_2_ solutions of various concentrations under the same ideal conditions at 652 nm, the quantity of H_2_O_2_ contained in the urine samples of the cancer patient is calculated from an existing calibration plot as shown in Figure 13 [5]. The results obtained by applying the percentage recovery formula are summarized in Table 2.

## 3. Materials and Methods

Analytical-grade chemicals such as cerium (III) nitrate hexahydrates. Ce(NO_3_)_3_.6H_2_O (99.9%), cobalt (II) nitrate hexahydrate, Co(NO_3_)_2._.6H_2_O (99.99%), KOH (99.9%), 3,3′,5,5′-tetramethylbenzidine (TMB), ethanol C_2_H_5_OH (99.8%), and activated carbon (99.9%) were supplied by Sigma-Aldrich. H_2_O_2_ (35%) was obtained from Merck KGaA. The phosphate buffer saline (PBS) in various pH ranges was purchased from Bio World. Disposable cuvettes of 2 mL were procured from Kartell. Different solutions having various concentrations were prepared in deionized water. All the above chemicals were used without any further need to purify them.

### 3.1. Instrumentation

Using the UV–Vis spectrophotometer model Shimadzu UV-1800, Germany absorption spectra were recorded. By using Fourier transform infrared spectroscopy (FTIR, MX 300, Mohrsville, PA, USA), the functional groups were studied over wavenumber range of 4000–500 cm^−1^. Furthermore, FTIR also detects C=C groups of carbon and vibrations of CeO_2_, which are related to bulk material rather than to functional groups. The surface morphology and size of the nanocomposite were examined by using a scanning electron microscope (SEM, JSM-IT 100 Jeol, Tokyo, Japan), and by using EDX (equipped with the SEM) the elemental analysis was performed. The accelerated voltage of 20 kV was applied to take images at different magnifications. The crystal structure of the Co/CeO_2_/C/IL was studied by using powder X-ray diffraction (XRD) and a Bruker Smart Apex CCD (Billerica, MA, USA) over the 2θ range of 20–80°. The Raman spectrum at room temperature was recorded by using the portable Raman instrument attached to a microscope (i-Raman B&W TEK Inc, Plainsboro, NJ, USA, 20× objectives).

### 3.2. Synthesis of Cobalt-Doped Cerium Oxide/Activated Carbon Nanocomposite

For the synthesis of cobalt-doped cerium oxide/activated carbon nanocomposite, precursors such as cerium (III) nitrate hexahydrate (Ce(NO_3_)_3_·6H_2_O) and cobalt (II) nitrate hexahydrate (Co(NO_3_)_2_·6H_2_O) were used, and the co-precipitating agent potassium hydroxide (KOH) was also used. The concentration of cerium precursor was 0.3 M, and that of Co precursor (Co(NO_3_)_2_·6H_2_O) doping was 0.01 M. At the start, 150 mL of 0.3 M cerium nitrate solution was prepared at room temperature and stirred vigorously. Then 150 mL of cobalt nitrate solution with 0.01 M concentrations was gently added to the cerium nitrate solution. Secondly, 1 M aqueous solution of KOH was added drop by drop to the reaction mixture until the pH value reached 11 and at the same time, 1.19 g of activated carbon was also added. Then the prepared colloidal solution was preserved in a flask for 2 days at room temperature for precipitation. Finally, by centrifuging at 6000 rpm, the precipitates were obtained and washed with ethanol and deionized water four times to eliminate the impurities. Then the prepared product was dried at 45 °C for 2 h to perform the characterization. The synthesis of the nanocomposite was repeated to confirm its reproducibility and obtain a good amount for further experimental work.

### 3.3. Capping of Co/CeO_2_/C Nanocomposite with Ionic Liquid

The capping of Co/CeO_2_/C with ionic liquid was carried out as follows: First, Co/CeO_2_/C (6 mg) was added to 1 mL of 1-Butyl-3-methylimidazolium tetrafluoroborate in a china dish and macerated the nanocomposite, and the ionic liquid for about 40 min with a mortar and pestle until a well dispersed blackish mixture of ionic liquid coated Co/CeO_2_/C was obtained. The mixture was kept in a sealed container for future applications [46,47,48].

### 3.4. Detection of Hydrogen Peroxide

The peroxidase-like activity of Co/CeO_2_/C/IL was evaluated for the determination of H_2_O_2_ by using TMB as a chromogenic substrate. In an Eppendorf tube, 50 μL dispersed ionic liquid nanocomposite, 500 μL of phosphate buffer saline (pH 7), and 180 μL of 10 mM TMB solutions were mixed, and finally, 85 μL H_2_O_2_ (6 mM) was added into the reaction mixture and incubated for 3 min under room temperature to detect the optical change. The color of the solution was changed from transparent to bluish-green. The UV–Vis spectrophotometer was used to record the absorption spectra of the resultant solution at 652 nm. Some experimental conditions, such as TMB concentration, time, pH, response amount of capped nanocomposite, and H_2_O_2_ concentration, were optimized to obtain the best performance.

## 4. Conclusions

The Co-precipitation method was used in the current study to synthesize Co-CeO_2_/C, which was then characterized by using a variety of techniques, including UV–Vis, FTIR, SEM, EDX analysis, Raman spectroscopic techniques, and X-ray diffraction. The H_2_O_2_ is detected via a colorimetric approach by using the peroxidase mimicking ionic liquid-capped nanocomposite to accelerate the oxidation of TMB in the presence of H_2_O_2_. The Co-CeO_2_/C nanocomposite electrical conductivity and enzyme mimicry capabilities are enhanced by the ionic liquid utilized in addition to stabilizing the nanomaterials. The proposed sensor exhibits a wide linear range of 1 × 10^−9^ M to 3.6 × 10^−7^ M, high sensitivity of 1.4 × 10^−8^ M, and a low limit of detection of 1.3 × 10^−8^ M under ideal conditions. The sensor shows a good result within 2 min and is highly selective and sensitive for H_2_O_2_ even in the presence of an interfering substance. In addition, the sensing probe was successfully used to find H_2_O_2_ in cancer patients’ urine samples.

## Data Availability

Not applicable.

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
