# Peer review of "Co-Doped CeO2/Activated C Nanocomposite Functionalized with Ionic Liquid for Colorimetric Biosensing of H2O2 via Peroxidase Mimicking"

_molecules, 2023, doi:10.3390/molecules28083325_

Round 1
Reviewer 1 Report
Overview:
In molecules-2295938 “Co doped CeO2/Activated C Nanocomposite functionalized with Ionic Liquid for Colorimetric Biosensing of H2O2 via Peroxidase Mimicking”, authors report on the formation of Co-doped CeO2/a-C composite allowing for H2O2 sensing in the presence of the ion liquid and chromogenic substrate. The selectivity and speed of the observed sensing response are of interest. However, the manuscript is not free of errors and substandard discussions and is in a dire need of serious revision before the final decision.
Major comments:
1) Authors report on the complex physical and chemical processes taking place in a complex media (analyte+TMB+IL+Co/CeO2/C). As for the studied IL-submerged Co-doped CeO2/a-C sample, the role of all of the components should be discussed. Were there any previous studies assessing the sensing performance of CeO2/Co-doped CeO2/amorphous carbon/CeO2+IL? What processes would happen in a less effective way if there will be no a-C/IL/Co doping? Currently manuscript provides no information about the choice of the components and the design of the composite. The role of ionic liquid in agglomeration prevention is presented in line 272 – it would be beneficial to create similar discussions about other components and present them in a single section.
2) Lines 228-230: “The two strong peaks at 1327 cm-1 and 1595 cm-1 related to D band and G band of carbonaceous samples were found to shift downward to cm-1 and 1160 cm-1 respectively.”: first of all, the positions of the D and G line are not “set in stone” at 1327 cm-1 and 1595 cm-1, as they can vary in 1320-1360 cm-1 and 1520-1600 cm-1 range [10.3390/magnetochemistry8120171]. Secondly, in Ref. [50] the shift of CeO2-related lines to 598 and 1167 cm-1 is discussed, but the shift of carbon-related D and G is not reported. For me, the positioning of D and G lines at 846 cm-1 and 1160 cm-1 seems highly unlikely (the ranges of their typical variation are 1320-1360 cm-1 and 1520-1600 cm-1). Therefore, this fragment should be revised and appropriate references should be provided to discuss the origin of the peaks.
3) You detect an sp-carbon-related line in FTIR spectrum at 2318 cm-1. However, you apparently don’t detect a Raman sp-carbon-related line at 2000-2400 cm-1 range. Could you discuss why? Raman and FTIR are complementary techniques, and their results are interconnected. I suppose that the FTIR “line” at 2318 cm-1 observed by authors is related to some instrumental effects rather than carbyne/sp-carbon/polyyne/cumulene.
4) Line 258-259, “The band gap energy of the nanocomposite is equal to the energy of the absorbed photon.”: appropriate references should be provided. Additionally, which component of the composite has a band gap of 1240/653=1.9 eV, is it affected by the defects and by other components of the composite?
5) Line 258-259, “The electron is excited from the valence band to the conduction band because electrons and H+ are produced due to the absorption of photons.”: do you mean that the dissociation of TMB or H2O2 takes place during this interaction? How is the energy transferred from the nanocomposite to the dissociated molecule (will the adsorbed molecule be dissociated more efficiently)? The sentence should be clarified to indicate what substructure is affected by this process and appropriate references should be provided.
6) Table 2 is especially confusing: first, what is "Detected"; second, what was the "percentage recovery formula"; third, what is RSD and how was it assessed? Fourth, what is the formula used for the "H2O2 found" calculation?
7) The design of the study associated with the Table 2 is also questionable: is H2O2 typically detected in the "urine samples taken from cancer patients"? If yes, why did you add more of it into the urine samples and how could the presence of patient-produced H2O2 affect the results of the assessment? If no, why was the studied analyte (real urine+H2O2) chosen? As H2O2 of a relatively high concentration was artificially added to the samples, I think that the claims “..was used effectively to detect H2O2 in cancer patients' urine samples…” (line 39-40) and “Also, the sensing probe was successfully used to find H2O2 in cancer patient’s urine samples.” (lines 405-406) are misleading and should be revised.
8) Line 22, “Both activated C and IL have a synergistic effect on the conductivity of the nanocomposites”: first, what kind of the conductivity do you mean? Did you assess it somehow? Second, please add the discussion in the text to support this claim, as it is currently presented only in the abstract.
Minor comments:
9) As indicated in [https://www.mdpi.com/journal/molecules/instructions], the abstract should be a total of about 200 words maximum. In your abstract, there is more than 300 words.
10) Lines 111-112: "By using Fourier transform Infrared (FTIR, MX 300 USA) the functional groups were studied over wave length range of 4000-500 cm-1": first, infrared _spectroscopy_; second, FTIR also detects C=C groups of carbon and vibrations of CeO2, which are related to bulk material rather than to functional groups; third, “x-y cm-1” is a wavenumber range rather than wavelength range.
11) Lines 124-125: “and that of 124 Co doping was 0.01 M.”: is it a concentration of Co or of Co-containing precursor?
12) Fig. 1: consider indicating the origin and/or position of major lines in the figure. Currently, the CeO-related lines at 500-900 cm-1 are not visible, consider showing an enlarged fragment of the spectra in this range.
13) In Ref. 41, the line at 23xx cm-1 is attributed to C≡C (polyyne chains, sp-hybridized carbon) rather than C=C. In Ref. 42, its assignment to C=C is probably related to cumulenic chain bonds (sp-hybridized carbon), as the C=C stretching vibrations line of sp2-carbon is located at 1560-1600 cm-1 [DOI: 10.1080/10408347.2016.1157013]. I think that the fragment at lines 168-169 “the peak relating to 2318 cm-1 indicates the presence of C=C groups [41,42]” should be revised to indicate that this peak is attributed to sp-carbon rather than sp2-carbon. Additionally, this peak is barely visible in the spectra, so I suggest that enlarged inset/sub-figure should be shown to prove it.
14) Line 167-168 “The peaks at 1495 cm-1 correlated to C-H symmetric and asymmetric bending vibration”: do you mean that there are multiple peaks in this wavenumber? The fragment should be revised.
16) Fig. 1: does a rightmost feature at 5xx cm-1 depicts the intensive line? If it is an instrumental feature, it shouldn’t be shown in the spectrum to avoid misleading the readers.
16) Line 178, “111, 200, 220, 311, 400 and 331”: Please indicate if this set of lines is attributed to CeO2, or it is related some other structure.
17) Fig. 2: blank spaces at 0-10° and 80-90° should be removed or reduced.
18) Line 207, could you please communicate on what does “(magnification of 60 HVx 1000)” mean?
19) Line 214, replace “EDX analysis” with “EDX spectrum” to distinguish the caption of the figure from the one of Table 1.
20) Table 1, in “Weight % Sigma” column, do you show the error of “Weight %”? I think the name of the column should be revised.
21) Table 1, column 1: can EDX really distinguish Co+ from Co? Appropriate discussion and references should be provided.
22) Lines 224-225, by “oxygen holes”, did you mean “oxygen vacancies”?
23) Figure 5: blank spaces at the right and left sides of the figure should be reduced. The origin of important peaks should be indicated. If no important peaks are located in 1500+ cm-1 range, this fragment of the spectra shouldn’t be shown.
2) Lines 232-233, “The D band is related to the disorderness or sp3 structural defect sites”: the D-line is attributed to the breathing mode of sp2-carbon rings, being active in the presence of disorder. Both D- and G-lines are related to graphitic structures, as the cross-section of the inelastic light scattering related to its interaction with sp2-clusters is 1–2 orders of magnitude higher than the one of the sp3 -carbon (see [10.1007/s00339-022-06062-2] and references within). The discussion should be revised.
25) Lines 234-235, “The peaks at 110, 118, 237 and 601 cm-1”: currently they are not visible in the picture, please make an inset supporting your discussion.
26) Lines 240-242, “In order to detect hydrogen peroxide using colorimetric approach a very simple method of carbon decorated ionic liquid coated cobalt doped cerium oxide nanocomposite was used.”: please revise the sentence, as currently it is hardly understandable.
27) Line 249: was the NCP abbreviation defined previously?
28) Fig. 6:
Were 85 μL of H2O2 added to both of the samples, as shown in line 249, or they were not added to the sample (A), as indicated in line 250? Please revise the caption.
29) Line 260-261, “H2O2 will also prevent H+ and electron from recombination.”: please explain the mechanism in more detail.
30) Lines 264-265, “the absorption capability of the nanocomposite in visible light ul- timately more OH radicals are produced”: please revise the sentence to make it more understandable.
31) Scheme 1: is it possible to communicate the role of Co/CeO2 /C/IL in the scheme? The quality of the right fragment showing ionized TMF is substandard.
32) In line 265, you claim that “more OH radicals are produced which enhance further oxidation of TMB”, however, the oxidation is not shown in the scheme 1 at all. Could you explain the discrepancy?
33) In Fig,7 and 8, is the showed absorbance is an absorbance measured at particular wavelength? Or is it a normalized intensity of the cumulative absorbance peak? Please supplement the discussion.
34) Lines 322-324: “because all the available Co-CeO2/C nanocomposite was used in the oxidation of TMB and further oxidation of TMB was very low”: isn’t the oxidation of the TMB connected to the H2O2 dissociation? Please revise the fragment.
35) Section 3.4.3 — is the observed performance peaking at 7.4 pH good or bad for practical applications? What can be concluded from this subsection?
36) Fig 7-11: why mark the points A-G, A-K? These points represent various sets of conditions for different figures, therefore, similar naming for various figures is confusing.
37) There is no need to add the fitting parameters to the inset of Fig.11B if you don't discuss them in the text.
38) Lines 396-397: "Fourier, FTIR" is redundant.
Reviewer 2 Report
In the present study, ionic liquid (IL) coated cobalt (Co) doped cerium oxide (CeO2)/activated carbon (C) nanocomposite has been used to assess the peroxidase-like activity for the colorimetric detection of H2O2.
The work is interesting, structure of the article is appropriate but the authors must address the following comments and incorporate the changes in the manuscript before it can be considered for publication.
- the title suggests that authors synthesise nanocomposit - there is no evidence in this work that synthesised Co doped CeO2/Activated C are indeed nanocomposite. (the average size obtained only by XRD analysis cannot be taken into account; AFM or TEM analyses is needed
Round 2
Reviewer 1 Report
Authors have addressed most of my comments and improved the manuscript. However, there are some remaining issues.
1) Line 17, does H2O2 really “causes different types of cancers”? As I understand, it is better to denote that its emergence is caused by the cancer.
2) In Fig.1(a,b), instrumental features at low wavenumbers (in the right parts of the figures) should be removed and the origin of most notable lines should be noted.
3) If you create a document with the supplementary figures, refer to them in the original text (see https://www.mdpi.com/journal/molecules/instructions, “Supplementary Materials”).
4) In Section 3.1.5, the origin of the lines located at 846 cm-1 and 1160 cm-1 is not investigated. However, the D- and G-lines of amorphous carbon are discussed, although they are apparently not present in the spectra. Please revise the discussion.
5) As a follow-up to previously-formulated question 8: please indicate that you mean electrical conductivity, not a thermal one (lines 21, 70 etc.).
6) Line 296, please revise the fragment “However the color became disappear”.
7) Lines 353-354, please indicate the measurement units of the intercept and slope.
Author Response
Reviewer 1 remaining comments and authors response
Authors have addressed most of my comments and improved the manuscript. However, there are some remaining issues.
Response: Worthy reviewer, the authors are grateful for your comments. We appreciate your time, guidance and contribution. It was a great learning experience.
1) Line 17, does H2O2 really “causes different types of cancers”? As I understand, it is better to denote that its emergence is caused by the cancer.
Response: Thank you for the comment. Indeed H2O2 is a complex analyte and this is what makes it interesting to study this analyte.
There are two broad types of situations related to H2O2; first one is regarding the role of H2O2 in cancer development. Miguel Lopez-Lazaro in his review in cancer letters in 2007 reviewed various studies and stated it in the following manner;
“It is known that H2O2 is associated with DNA damage, mutations and genetic instability Park et al., 2005; Hunt et al., 1998; Jackson and Loeb 2000; Pericone et al., 2002; Henle and Linn 1997. H2O2 induced DNA damage seems to be mediated by OH radical generated from H2O2 by Fenton reaction Park et al., 2005; Henle and Linn 1997; Imlay and Linn 1988. Several studies have also demonstrated that H2O2 can induce cell proliferation (Burdon, 1995; Zanetti et al., 2002 apoptosis resistance Brown et al., 1999; Bello et al., 1999, increased angiogenesis Qian et al., 2003; Arbiser et al., 2002, and invasion and metastasis Nelson et al., 2003; Nishikawa et al., 2004”.
Similarly in a recent paper from our group Nishan et al., 2023 https://doi.org/10.1007/s13369-023-07791-z it is mentioned “Although H2O2 is considered to be less toxic, it can easily damage numerous biomolecules including lipids, proteins, carbohydrates, etc. In the body, it is linked to the incidence of various clinical signs, and excess H2O2 can trigger gene mutation, inflammation, cancer, diabetes mellitus, etc Science 312, 1882–1883 (2006); Biosens. Bioelectron. 87, 101–107 (2017); Acc. Chem.Res. 44, 793–804 (2011).
The second situation is during the cancer which we stated in our previous comments;
“It is well documented that the urinary excretion of H2O2 in cancer patients is two- to three fold higher than the normal individuals Banerjee et al. 2003a Clin Chim Acta 334:205–9”.
2) In Fig.1(a,b), instrumental features at low wavenumbers (in the right parts of the figures) should be removed and the origin of most notable lines should be noted.
Response: Thank you for the comment. The instrumental features removed and important features of the spectra highlighted in Fig.1(a,b) as can be seen in the revised manuscript.
3) If you create a document with the supplementary figures, refer to them in the original text (see https://www.mdpi.com/journal/molecules/instructions, “Supplementary Materials”).
Response: Thank you for the comment. The supplementary figure S1 renamed according to the journal format.
4) In Section 3.1.5, the origin of the lines located at 846 cm-1 and 1160 cm-1 is not investigated. However, the D- and G-lines of amorphous carbon are discussed, although they are apparently not present in the spectra. Please revise the discussion.
Response: Thank you for the comment. We agree that the following information were irrelevant and confusing here removed;
This information added for the kind consideration of the worthy reviewer;
The band at 846 cm-1 corresponds to the Ñ´(O–O) stretching vibration of peroxo species formed on CeO2, While The bands at 1160 cm–1 were attributed to second-order longitudinal optical (2LO) mode [36].
5) As a follow-up to previously-formulated question 8: please indicate that you mean electrical conductivity, not a thermal one (lines 21, 70 etc.).
Response: Thank you for the comment. The word electrical added to conductivity for more clarity.
6) Line 296, please revise the fragment “However the color became disappear”.
Response: Thank you for the comment. The fragment revised as suggested.
7) Lines 353-354, please indicate the measurement units of the intercept and slope.
Response: Units of the intercept and slope mentioned as can be seen in the revised manuscript.
